# Association of cognitive reserve with 9-year domain-specific cognitive trajectories and risk of cognitive impairment in Mexican older adults

Martina Ferrari-Díaz[1]*, Ashuin Kammar-García[2], Juan Silva-Pereyra[3]*, Carmen García-Peña[4]

1 Facultad de Humanidades y Ciencias Sociales, Universidad La Salle México, Mexico City, Mexico, 2 Dirección de Investigación, Instituto Nacional de Geriatría, Mexico City, Mexico, 3 Facultad de Estudios Superiores Iztacala, Universidad Nacional Autónoma de México, Tlalnepantla, Mexico, 4 Dirección General, Instituto Nacional de Geriatría, Mexico City, Mexico

* martinaferraridiaz@gmail.com (MFD); jsilvapereyra@gmail.com (JSP)

## Abstract

Cognitive reserve (CR) refers to the adaptation of cognitive performance to endure brain pathology or the aging process. CR can be categorized into static (education and occupation) or dynamic (leisure and physical activities) proxies. Typically, longitudinal studies assess CR as a composite score at baseline and cognitive performance as a global score. This study aimed to compare the relationship between different CR proxies (static and dynamic) with 9-year domain-specific cognitive trajectories, and the risk of cognitive impairment in older adults. Data from the latest four waves of the Mexican Health and Aging Study (MHAS; n = 3102, baseline mean age = 66.62 years) were used. Mixed effects models were performed with CR as independent variables and cognitive trajectories (verbal memory encoding and retrieval, verbal fluency, constructional praxis, visual attention, and memory) as outcomes. Education and leisure activities were significant positive predictors of all cognitive domains. Physical activities were a positive predictor of verbal fluency and verbal memory encoding only. Occupation was a positive predictor of verbal fluency and visual attention. Logistic regression analysis was performed to assess the relationship between CR and the risk of cognitive impairment, where education (OR: 0.79, 95% CI: 0.76, 0.83), occupational complexity (OR: 0.85, 95% CI: 0.77, 0.95), and leisure activities (OR: 0.96, 95% CI: 0.95, 0.97) were significant protective factors. Increasing the years of education can serve as a preventive strategy to delay the clinical manifestation of cognitive impairment while implementing leisure activities can act as an intervention to promote cognition even in later years.

**Data availability statement:** Data requests can be submitted online at http://www.mhasweb.org/.

**Funding:** The MHAS (Mexican Health and Aging Study) is partly sponsored by the National Institutes of Health/National Institute on Aging (grant number NIH R01AG018016) and the INEGI (Instituto Nacional de Estadística y Geografía) in Mexico. Data files and documentation are for public use and available at www.MHASweb.org. The funders had no role in study design, data collection and analysis, decision to publish, or preparation of the manuscript. There was no additional external funding received for this study.

**Competing interests:** The authors have declared that no competing interests exist.

# 1 Introduction

Cognitive reserve (CR) is defined as the adaptation of cognitive performance to endure brain pathology, lesions, or the aging process [1]. It has been highly reported that CR positively influences abilities such as memory, verbal skills, and executive functions [2], particularly in healthy older adults and those at risk of dementia.

Most of the studies assess CR and cognitive performance at a single point in time using cross-sectional designs. However, this methodology does not allow for the evaluation of changes over time in lifestyle factors on cognitive performance and it is impossible to identify a causal relationship. In principle, longitudinal studies would allow us to better understand the effect of CR on cognitive trajectories. However, there are still mixed findings about the effects of CR on longitudinal cognitive trajectories [3]. A review [3] and recent studies reported that higher scores of CR relate to better cognitive performance at baseline [4–7] and lower risk of dementia [8,9]. But when it comes to cognitive trajectories or rates of cognitive impairment (changes in cognitive performance over time), the findings are less clear. According to the review from Pettigrew and Soldan [3], some studies reported that higher scores of CR correlate with slower rates of cognitive impairment, while other studies found faster rates of cognitive impairment associated with higher scores of CR, and some studies reported no relationship with CR and the rate of cognitive impairment even if they found differences at baseline [3,7].

These mixed findings may be attributable to the characteristics of the sample, methodological issues, or the operationalizations of variables [3]. The diverse populations studied (Chinese, American, English, etc.) may influence the results because of social, economic, and cultural differences. Another important factor is the cognitive status of the sample at baseline, some studies included only healthy older adults [4,5,8,9], meanwhile, others also considered participants with Mild Cognitive Impairment (MCI) and dementia [6,10]. Furthermore, the several follow-up times and waves of assessment may impact the evolution of the characteristics of the sample. For instance, the theoretical explanation proposed by Stern [11] suggests that Alzheimer's Disease (AD) patients with higher reserve can tolerate more pathology before exhibiting clinical symptoms, however, when symptomatology starts, they may present a faster cognitive decline because neuropathology is more advanced. Including participants with different cognitive statuses from the baseline may provide a wider understanding of the mechanism of CR in both normal and pathological aging.

However, the variable that may account for these inconsistencies is the operationalization of CR, which remains a challenge. It is a construct that is not directly measurable, and proxy measures of CR are used to indirectly assess it. Sociobehavioral proxies related to CR are years of education, occupation, and leisure activities, measured by mental, physical and social activities [12], and according to Malek-Ahmadi et al. [13] can be categorized into static or dynamic. Static measures are stable over time, such as the level of education or years of education; however, this proxy does not account for changes through life and may not reflect the quality of the education [3]. Dynamic measures are characterized by their changing nature, such as verbal

intelligence and engagement in cognitive activities [13]. Some longitudinal studies used proxy variables of CR as separate predictors, such as vocabulary, and years of education [7]. Most studies used composite scores to assess CR through the combination of different proxy variables like reading level, vocabulary, occupation complexity, education, marital status, engagement in social, cognitive, and physical activities [4–6,8–10]. Another difficulty in the operationalization of CR, is that although some proxy measures of CR in longitudinal studies could reflect dynamic changes, they are frequently used as static indicators by focusing solely on baseline scores. As a result, these studies overlook possible fluctuations in CR that might correlate with dementia progression [14].

Using a composite score to assess CR considering a set of variables may condensate the ample variety of activities and lifestyle factors that a person experienced throughout their life; however, it may prevent us from distinguishing the effect of each different proxy in specific cognitive domains [2,15]. It has been proposed that different proxies of CR are related to different cognitive domains: dynamic proxies are associated with fluid cognitive abilities and static proxies with crystallized cognitive abilities [13,16]. For example, studies using dynamic proxies such as leisure and social activities, and verbal intelligence have found associations with higher spatial working memory, perceptual organization, processing speed, and executive functions [17,18]. In contrast, static proxies, like educational level, have been linked to higher performance in attention, verbal memory and orientation [19]. For this reason, relying solely on a composite score of CR and a global cognition measure may cover the specific contribution of individual proxy variables in explaining domain-specific cognitive performance and trajectories. Understanding these distinct associations could help design targeted intervention strategies [6], not only for pathological aging but also to enhance cognitive functioning during normal aging.

Therefore, the aim of this study is to compare the relationship of different longitudinal proxies of CR (static, dynamic, and a combination of both) with 9-year domain-specific cognitive trajectories, and the risk of cognitive impairment in Mexican older adults.

Our hypotheses are that: a) dynamic proxies of CR will be associated with higher cognitive performance trajectories of constructional praxis and verbal fluency performance, b) static proxies of CR will be associated with higher cognitive performance trajectories of attention, verbal and visual memory performance, c) the combination of both static and dynamic proxies of CR will be a better predictor of a lower risk of cognitive impairment than using each proxy of CR separately.

## 2 Materials and methods

### 2.1 Study design and sample selection

Data were retrieved from the Mexican Health and Aging Study (MHAS) [20] on July 5, 2023. The authors did not have access to personally identifiable information of the participants. MHAS is a nationally representative longitudinal study of Mexicans aged 50 years or older from all 32 states of the country in urban and rural areas. The baseline survey was conducted in 2001 with seven follow-up interviews until 2021. Ethical approval was granted by the Institutional Review Board or Ethics Committee of the University of Texas Medical Branch, the INEGI in Mexico, and the Instituto Nacional de Salud Pública (INSP) in Mexico, detailed information can be found on the original survey.

For the present retrospective cohort study, we used data from the latest four waves that included the assessment of cognitive performance: 2012, 2015, 2018, and 2021 [21]. We included participants aged 60 years and over. We excluded participants with missing data on CR or cognitive performance variables in any of the four waves. Finally, we merged the four databases to create a sample of participants with complete information. Fig 1 depicts the flowchart of the baseline sample selection.

### 2.2 Independent variables

To evaluate static and dynamic proxies of CR we considered a set of questions and variables assessed in the MHAS. For each proxy, we considered the score obtained in every wave assessed: from baseline (2012) to 2021. We searched for

**Fig 1. Flowchart of the baseline sample selection.**

differences in the CR proxies throughout the waves to confirm if they could be considered static or dynamic. Only education was considered a static proxy of CR, while occupation, leisure and physical activities were considered dynamic proxies. According to previous studies, occupation is typically considered a static proxy of CR [13]. However, in the present study, we observed that occupational complexity varied over time. Therefore, we analyzed it as a dynamic proxy of CR.

**2.2.1 Education.** We considered education in years reported in the MHAS.

**2.2.2 Occupational complexity.** We used a question of the MHAS regarding type of occupation, in the original study the answers were categorized according to the Mexican classification of occupations of the National Institute of Statistics and Geography of Mexico (INEGI) into occupational types. For the present study, we recoded the classification using the five levels of employment proposed by Nucci and colleagues [22] in the Cognitive Reserve Index questionnaire (CRIq) which are differentiated by the cognitive input and the level of responsibility required. The first one, is the unskilled manual work (farm worker, gardener, driver, waiter, electrician, housemaid, etc.); the second one is the skilled manual work (like a craftsperson, cook, tailor, nurse, hairdresser, etc.); the third one is the skilled non-manual or technical work (like a business owner, an employee, an agent [sales or estate], a musician, a religious worker, etc.); the fourth one is the professional occupation (like a managing director of a small company, a teacher, a doctor, a lawyer, a psychologist, an engineer, etc.); the last and fifth one is the highly intellectual occupation (like managing director of a large company, judge, scientific professional, university professor, etc.). We completed missing data retrieving information from the SimpleMHAS (a dataset with reduced selection of variables) which only specified if they were working (the full MHAS database specify the classification of occupations), unemployed, if they did household chores, or they were currently looking for a job. The information changed throughout the waves, for this study we considered the occupation they reported for each wave (and then recoded as described below), and if they were unemployed or looking for a job, we considered the lowest score.

Occupational complexity data was quantified on a scale from 1 to 5 according to the level previously described.

**2.2.3 Leisure activities.** We retrieved a set of 12 questions from the MHAS regarding the frequency of time used in different kinds of activities such as: caring for a sick/disabled adult, or a child, volunteering, attending a social club, communicating with family and friends, reading, playing games, doing puzzles, watching TV, crafting, home maintenance, and training courses. Volunteering was included in this proxy and not in occupation because of the usual definition of leisure activities as voluntary activities associated with motivational factors [23].

We considered the total sum of the frequency of time used in these activities reported by the participants (weekly or monthly). We calculated the daily frequency in a month for each activity, being 360 the maximum possible amount for this variable.

**2.2.4 Physical activities.** We considered the only question from the MHAS regarding physical activity which asked if they took part in exercise or hard physical work three times or more per week in a row, according to their answers we classified the participants as physically active or physically inactive.

## 2.3 Dependent variables

**2.3.1 Domain-specific cognitive trajectories.** To evaluate domain-specific cognitive trajectories, we considered a set of cognitive tasks assessed in the MHAS. We retrieved five cognitive tasks associated with different cognitive abilities: memory (verbal and visual), constructional praxis, verbal fluency, and attention. Verbal memory was measured using the number of words recalled (both immediate and delayed recall) from a list of eight words. Constructional praxis was measured through the copy of a figure composed of three parts, the global score ranged from 0 to 6, assigning 0–2 points for each part (1 point if it is present and 1 point if it is positioned correctly). Visual memory was measured through the ability to recall the three parts of the figure previously copied. Semantic verbal fluency was measured through the number of animals recalled in one minute. Selective attention was measured through the number of figures (60 targets) identified in one minute using a visual scanning task.

We used the raw scores obtained in each of the six tasks during the four waves: verbal memory (encoding and retrieval), constructional praxis, visual memory, semantic verbal fluency, and visual attention.

**2.3.2 Risk of cognitive impairment.** We classified the participants into two categories: the normal cognition group or the cognitively impaired group.

We first computed the participants' global cognitive performance using the sum of all the raw scores. In the case of verbal fluency, instead of the number of animals recalled we used a coded score proposed by Mejía-Arango and colleagues [24], where 0–8 animals were coded as 1 point, 9–18 as 2 points, 19–24 as 3 points, and 25–50 animals as 4 points. Then, we added the orientation score (from 0 to 3 points), even if it was not included in the domain-specific cognitive trajectories analysis, because it is considered an important measure to detect cognitive impairment.

To classify the sample, we adapted a categorization system proposed by Michaels-Obregón and colleagues [25] to achieve the objectives of the present study. The measures of global cognitive score (maximum of 95) were transformed into z-scores within three age groups (60–69, 70–79, and 80 years or more). According to their global z-score the participants were classified into normal (z-score of −1.5 or more) or with cognitive impairment (z-score of −1.5 or less). Even if there is no consensus about the cutoff scores, individuals with MCI are typically 1 to 1.5 standard deviations below their counterparts [26], however, in different recent studies a z-score of −1.5 or less is used as an indicator of cognitive impairment, no dementia (CIND) or to define the presence of cognitive impairment [26–29].

## 2.4 Covariates

We considered age (years), sex, comorbidities, depressive symptoms, and socioeconomic status in each wave as covariates for the statistical analysis. Comorbidities were assessed through the sum of twelve different types of diseases (hypertension, diabetes, cancer, respiratory illness, heart condition, stroke, arthritis/rheumatism, liver infection, kidney infection, tuberculosis, pneumonia, herpes/zoster) reported by participants and their caregivers, although self-reported, the questionnaire specifies that the condition must have been diagnosed by a doctor or medical personnel. Depressive symptoms were assessed through the sum of nine questions regarding their feelings (e.g., depression, happiness, sadness, etc.) during the last week. Socioeconomic status was assessed using the answer to a self-reported financial situation question, the answers ranged from 1 to 5 ("poor", "fair", "good", "very good", and "excellent") which were used as a quantification for this variable.

## 2.5 Statistical analysis

The data will be described as mean and standard deviation (SD) for quantitative variables and as frequency and percentage for categorical variables.

To determine the effect of the different CR proxies (both static and dynamic) on cognitive trajectories, we used a linear mixed effects models using the year of evaluation as a random effect with a random intercept for every subject, every CR proxy was used as independent variables and cognitive trajectories (verbal memory encoding and retrieval, verbal fluency, constructional praxis, visual attention, and memory) as outcomes. We performed several multivariate models in which sex, age, comorbidities, depressive symptoms and socioeconomic status were included as covariates.

Logistic regression analysis was performed to evaluate the risk of cognitive impairment (normal cognition group vs cognitive impaired group) associated with the different CR proxies, considering sex, age, comorbidities, depressive symptoms, and socioeconomic status as covariates.

A multivariable logistic mixed effects model was performed to evaluate the risk of cognitive impairment (normal cognition group or cognitive impaired group) associated with the different CR proxies, considering sex, age, comorbidities, depressive symptoms, and socioeconomic status as covariates, the year of evaluation was used as random effect, and we calculated a random intercept for every subject.

The validity of the models was assessed through residual analysis. All statistical analyses were performed in R version 4.4.2 using the lme4 and lemerTest libraries.

# 3 Results

## 3.1 Sociodemographic characteristics

The descriptive characteristics of the total sample in the four waves are presented in Table 1. The final sample included 3102 participants (1766 male). Regarding the covariates, comorbidities increased through the years, while depressive symptoms and self-reported financial situation decreased. Except for years of education, the other CR proxies differed along the waves, even occupational complexity, which is theoretically considered a static proxy. In some cognitive domains, there was a mild increase in performance from wave 1 to wave 2, presenting a later decline throughout waves 3 and 4. This phenomenon was also observed in the percentage of cognitively impaired participants across the waves, which increased from wave 1 to waves 2 and 3, but unexpectedly decreased in wave 4. A notable proportion of participants reverted from the cognitively impaired group to the normal group across the waves, with 3.81% (n = 111) between 2012 and 2015, 3.97% (n = 115) between 2015 and 2018, and 4.78% (n = 139) between 2018 and 2021.

However, the mean sum of global cognitive performance presented a clear decreasing trend over the nine years.

## 3.2 CR and cognitive trajectories

The results of the mixed-effects models using the CR proxies (static and dynamic) as predictors and domain-specific cognitive trajectories as outcomes can be observed in Table 2.

Years of education were a positive predictor throughout the years for verbal fluency ($\beta = 0.24$, 95% CI = 0.22–0.27), visual attention ($\beta = 0.99$, 95% CI = 0.94–1.05), verbal memory encoding ($\beta = 0.06$, 95 CI = 0.06–0.07), verbal memory

Table 1. Descriptive characteristics of the sample by year.

| | Wave 1 (2012) n = 3102 | Wave 2 (2015) n = 3102 | Wave 3 (2018) n = 3102 | Wave 4 (2021) n = 3102 |
|---|---|---|---|---|
| **Covariates** | | | | |
| Age | 66.62 (5.18) | 69.46 (5.32) | 72.54 (5.34) | 75.58 (5.35) |
| Comorbidities | 1.07 (1.03) | 1.21 (1.10) | 1.27 (1.12) | 1.31 (1.12) |
| Depressive symptoms | 4.36 (1.77) | 4.34 (1.73) | 4.29 (1.75) | 4.23 (1.79) |
| Socioeconomic status | 3.86 (0.63) | 3.83 (0.63) | 3.78 (0.65) | 3.67 (0.68) |
| **Cognitive Reserve** | | | | |
| Years of education | 5.76 (4.45) | 5.76 (4.45) | 5.76 (4.45) | 5.76 (4.45) |
| Leisure activities | 21.08 (12.88) | 22.20 (12.01) | 19.70 (10.81) | 19.25 (10.79) |
| Physical activities | 0.41 (0.49) | 0.39 (0.49) | 0.29 (0.46) | 0.23 (0.42) |
| Occupational complexity | 0.88 (1.07) | 1.32 (1.17) | 1.18 (1.11) | 0.95 (1.12) |
| **Domain-specific cognitive trajectories** | | | | |
| Verbal fluency | 15.54 (4.78) | 16.09 (4.84) | 15.44 (4.87) | 14.64 (4.92) |
| Visual attention | 28.96 (14.04) | 28.67 (14.24) | 27.37 (14.60) | 24.56 (13.93) |
| Verbal memory encoding | 4.67 (1.54) | 4.91 (1.08) | 4.81 (1.12) | 4.52 (1.50) |
| Verbal memory retrieval | 4.43 (2.12) | 4.38 (2.01) | 4.28 (1.93) | 4.10 (1.93) |
| Constructional praxis | 5.62 (0.95) | 5.65 (0.82) | 5.58 (0.87) | 5.52 (1.06) |
| Visual memory | 4.95 (1.50) | 4.94 (1.53) | 4.84 (1.50) | 4.86 (1.52) |
| **Risk of cognitive impairment** | | | | |
| Sum of global cognition | 53.47 (16.60) | 53.41 (16.84) | 51.91 (17.13) | 48.26 (16.68) |
| Cognitively impaired | 182 (5.86%) | 193 (6.22%) | 205 (6,60%) | 187 (6.03%) |

**Table 2. Mixed effects models with cognitive trajectories as outcomes.**

| | Model 1 | Model 2[a] | Model 3[b] |
|---|---|---|---|
| | β (95% CI) | β (95% CI) | β (95% CI) |
| **Verbal fluency** | | | |
| Years of education | 0.29 (0.27-0.32)*** | 0.26 (0.24-0.28)*** | 0.24 (0.22-0.27)*** |
| Occupational complexity | 0.23 (0.16-0.30)*** | 0.21 (0.14-0.28)*** | 0.21 (0.18-0.24)*** |
| Leisure activities | 0.05 (0.05-0.06)*** | 0.06 (0.05-0.06)*** | 0.05 (0.04-0.05)*** |
| Physical activities | 0.50 (0.34-0.66)*** | 0.42 (0.26-0.59)*** | 0.31 (0.09-0.54)*** |
| **Visual attention** | | | |
| Years of education | 1.08 (1.02-1.15)*** | 1.03 (0.96-1.09)*** | 0.99 (0.94-1.05)*** |
| Occupational complexity | 0.38 (0.20-0.55)*** | 0.28 (0.10-0.46)** | 0.28 (0.09-0.46)** |
| Leisure activities | 0.11 (0.10-0.13)*** | 0.12 (0.10-0.13)*** | 0.11 (0.09-0.12)*** |
| Physical activities | 0.52 (0.12-0.91)** | 0.40 (0.01-0.79)* | 0.17 (−0.07-0.42) |
| **Verbal memory encoding** | | | |
| Years of education | 0.07 (0.06-0.08)*** | 0.07 (0.06-0.07)*** | 0.06 (0.06-0.07)*** |
| Occupational complexity | 0.04 (0.02-0.06)*** | 0.01 (−0.001-0.03) | 0.01 (−0.003-0.03) |
| Leisure activities | 0.02 (0.01-0.02)*** | 0.01 (0.01-0.02)*** | 0.009 (0.008-0.01)*** |
| Physical activities | 0.08 (0.03-0.12)** | 0.10 (0.03-0.18)*** | 0.07 (0.02-0.13)** |
| **Verbal memory retrieval** | | | |
| Years of education | 0.08 (0.07-0.09)*** | 0.08 (0.07-0.09)*** | 0.07 (0.06-0.08)*** |
| Occupational complexity | 0.06 (0.03-0.08)*** | 0.01 (−0.001-0.02) | 0.01 (0.00-0.03) |
| Leisure activities | 0.02 (0.01-0.02)*** | 0.01 (0.01-0.02)*** | 0.009 (0.007-0.01)*** |
| Physical activities | 0.06 (0.00-0.10) | 0.09 (0.01-0.17)** | 0.06 (0.00-0.12) |
| **Constructional praxis** | | | |
| Years of education | 0.04 (0.04-0.05)*** | 0.04 (0.03-0.04)*** | 0.03 (0.03-0.04)*** |
| Occupational complexity | 0.02 (0.008-0.02)* | 0.01 (0.00-0.02) | 0.01 (−0.002-0.03) |
| Leisure activities | 0.007 (0.005-0.009)*** | 0.007 (0.005-0.009)*** | 0.005 (0.004-0.006)*** |
| Physical activities | 0.03 (0.00-0.05) | 0.007 (−0.01-0.03) | −0.008 (−0.04-0.03) |
| **Visual memory** | | | |
| Years of education | 0.06 (0.06-0.07)*** | 0.06 (0.05-0.06)*** | 0.05 (0.04-0.06)*** |
| Occupational complexity | 0.03 (0.01-0.04)* | 0.01 (−0.003-0.03) | 0.01 (−0.01-0.04) |
| Leisure activities | 0.01 (0.008-0.01)*** | 0.01 (0.008-0.01)*** | 0.008 (0.005-0.01)*** |
| Physical activities | 0.07 (0.02-0.12)** | 0.04 (−0.002-0.07) | 0.01 (−0.04-0.07) |

[a]Adjusted model considering sex, age, comorbidities, depressive symptoms, and socioeconomic status as covariates.

[b]Multivariate model including all CR proxies and considering sex, age, comorbidities, depressive symptoms, and socioeconomic status as covariates.

**Note:** * p ≤ 0.05; ** p ≤ 0.01; *** p ≤ 0.001.

retrieval (β = 0.07, 95% CI = 0.06–0.08), constructional praxis (β = 0.03, 95% CI = 0.03–0.04), and visual memory (β = 0.05, 95% CI = 0.04–0.06) even when including the covariates and all CR proxies in the models.

Occupational complexity was a positive predictor through the follow-up for verbal fluency (β = 0.21, 95% CI = 0.18–0.24) and visual attention (β = 0.28, 95% CI = 0.09–0.46) even when adjusting for the covariates and in the model that included all CR proxies.

Leisure activities were a positive predictor along the years for verbal fluency (β = 0.05, 95% CI = 0.04–0.05), visual attention (β = 0.11, 95% CI = 0.09–0.12), verbal memory encoding (β = 0.009, 95 CI = 0.007–0.01), verbal memory retrieval

(β = 0.01, 95% CI = 0.01–0.02), constructional praxis (β = 0.005, 95% CI = 0.004–0.006), and visual memory (β = 0.008, 95% CI = 0.005–0.01) even when including the covariates and all CR proxies in the models.

Physical activities were a positive predictor during the 9-year trajectory for verbal fluency (β = 0.31, 95% CI = 0.09–0.54), and verbal memory encoding (β = 0.07, 95% CI = 0.02–0.13), even when adjusting for the covariates and including the other proxies of CR; and for visual attention (β = 0.40, 95% CI = 0.01–0.79) and verbal memory retrieval (β = 0.09, 95% CI = 0.01–0.17) only when adjusting for the covariates but not when including all CR proxies.

### 3.3 CR and risk of cognitive impairment

The results of the logistic regression analysis using the CR proxies (static and dynamic) as predictors for the risk of cognitive impairment can be observed in Table 3. Years of education (OR= 0.79, 95% CI = 0.76–0.83), leisure activities (OR= 0.96, 95% CI = 0.95–0.97), and occupational complexity (OR= 0.85, 95% CI = 0.77–0.95) were protective factors through the 9-year follow-up for the risk of cognitive impairment when including the covariates and all CR proxies in the models. However, occupational complexity (OR= 0.90, 95% CI = 0.80–1.01) was not a significant predictor of a lower risk of cognitive impairment when using the proxy of CR separately.

## 4 Discussion

Our first hypothesis was that dynamic proxies of CR would be associated with higher cognitive trajectories of constructional praxis and verbal fluency performance as reported by some studies [18,30]. Our study found that leisure activities were a significant positive predictor of all domain-specific cognitive trajectories (verbal fluency, visual attention, verbal memory encoding, verbal memory retrieval, constructional praxis, and visual memory), as demonstrated by Zhu et al. [31], where the performance of cognitive activities reported a greater effect than physical activities, and by Sanz Simon and colleagues [32] where engagement in leisure activities related to less decline in reasoning, speed, memory and better performance in vocabulary.

However, when it comes to another dynamic proxy of CR, the influence is less general, we found that physical activities were positive predictors only for verbal fluency and verbal memory (immediate recall). This result is consistent with those of a recent meta-analysis [33] which concluded that there is a relationship between physical activities and cognitive performance, but the association is weak. The studies included in the meta-analysis found a particular relationship with the same specific cognitive domains (memory and verbal fluency) of the present study. Additionally, it is important to highlight that physical activities were also significant positive predictors of visual attention and verbal memory (delayed recall) but only when other proxies of CR were not included. A systematic review and meta-analysis on physical activities and the prevention of cognitive decline and dementia found that studies adjusting for more confounders reported smaller effects [34]. This may explain our findings, as the inclusion of diverse confounding variables might better account for the cognitive trajectories observed in our sample.

**Table 3. Mixed Effects logistic regression analysis for risk of cognitive impairment.**

|  | Model 1 | Model 2[a] | Model 3[b] |
|---|---|---|---|
|  | OR (95% CI) | OR (95% CI) | OR (95% CI) |
| Years of education | 0.83 (0.76, 0.90)*** | 0.79 (0.74, 0.83)*** | 0.79 (0.76-0.83)*** |
| Occupational complexity | 0.84 (0.75, 0.95)** | 0.90 (0.80, 1.01) | 0.85 (0.77-0.95)** |
| Leisure activities | 0.99 (0.97, 0.99)* | 0.96 (0.94, 0.97)*** | 0.96 (0.95-0.97)*** |
| Physical activities | 1.14 (0.88, 1.47) | 1.03 (0.79, 1.33) | 1.11 (0.88-1.39) |

[a]Adjusted model considering sex, age, comorbidities, depressive symptoms, and socioeconomic status as covariates.

[b]Multivariate model including all CR proxies and considering sex, age, comorbidities, depressive symptoms, and socioeconomic status as covariates.

**Note:** * p ≤ 0.05; ** p ≤ 0.01; *** p ≤ 0.001.

Even if leisure and physical activities are both dynamic proxies of CR, the differences in the relationship between them and the cognitive outcomes may also be explained through their brain mechanisms. For instance, physical activities appear to help increase cerebral blood flow, neurotransmitters, growth factors, testosterone, and decreased insulin resistance [35] which promote a healthy status for the brain but do not directly lead to additional or more consolidated neural connections. Leisure activities, on the other hand, seem to enhance network efficiency and plasticity of neural circuits, and additionally, can foster stress management and emotional support [32]. Having a healthy physical condition is relevant for functionality, but leisure activities may impact several spheres related to a more global healthy aging: psychological (affective states, resilience, cognition), social (interactions, and social resources), behavioral (decision making, habits, etc.) and biological (endocrine, immune and central nervous system) [36].

Occupational complexity, considered in the present study as a dynamic proxy, was only a significant positive predictor of verbal fluency and visual attention. This result is concordant with a study by Finkel et al. [37], which concluded that occupational complexity related to a better verbal ability before retirement, and the one by Pool and colleagues [38] that found associations between occupational cognitive requirements and a slower rate of cognitive decline. Some studies considered the kind of occupational complexity (with people, with data, or precision) as features that may explain the domain-specific cognitive trajectories [37,39]. Additionally, in some countries like Mexico, disadvantaged people are more prone to work at jobs that require heavy physical labor and safety hazards, which are variables that may influence negatively their health status [40] and therefore their cognitive performance. A relevant finding regarding the characteristics of the sample was occupational mobility (both upward and downward along the waves). According to Malek-Ahmadi et al. [13], occupation is a static proxy of CR, however, in Mexico, even older adults experience occupational mobility because of socioeconomic conditions and the inequality of job opportunities [41]. Our second hypothesis proposed that static proxies of CR would be associated with higher cognitive trajectories in attention, verbal memory, and visual memory performance. We found that years of education significantly and positively predicted all cognitive trajectories assessed in the present study, which aligns with previous findings from cross-sectional research [2]. As noted in the introduction, the influence of CR on cognitive trajectories in longitudinal studies is less clear [3]. For instance, a systematic review and meta-analysis concluded that current evidence is insufficient to establish a consistent and strong relationship between education and changes in cognitive performance over time [42]. While our findings contrast with this conclusion, they are consistent with a recent study reporting that higher education predicts the maintenance of high cognitive functioning [43]. Notably, in our sample, years of education were associated with domain-specific cognitive trajectories despite the low average years of education (M = 5.76, S.D. = 4.45), highlighting the importance of this variable even in populations with limited formal education.

Our last hypothesis was that the combination of both static and dynamic proxies of CR would be a better predictor of a lower risk of cognitive impairment than using each proxy of CR separately. This study found that years of education, occupational complexity, and leisure activities were protective factors against global cognitive impairment when considering all CR proxies, which is consistent with previous literature. Lövden and colleagues [44], in a systematic review, reported that low educational attainment is associated with a greater incidence of dementia at any age. A recent study reported that occupational complexity, particularly jobs that require working with people, is related to a reduced risk of dementia [45]. A systematic review and meta-analysis by Yates et al. [46] concluded that participation in cognitively stimulating leisure activities is related to a reduced risk of dementia. Similarly, in a recent systematic review and meta-analysis, Liu and colleagues [47] found that high early CR, as measured by years of education, was associated with an 18% decreased risk of dementia, while mid-life CR, assessed by occupational complexity and social network, was linked to a 9% decreased risk and, finally, late-life CR, evaluated by cognitive activity and social connection, was related to a 19% decreased risk of dementia.

Interestingly, we found that occupational complexity was a positive predictor of a lower risk of cognitive impairment only when all proxies of CR were included, but not when considered individually. This finding may be explained by a

potentiation of the effect when considering other variables. As reported by Andel et al. [48], they identified a compensatory effect between leisure activities and occupational complexity, suggesting that individuals in occupations that impose few cognitive challenges can mitigate cognitive decline by engaging in leisure activities, and vice versa. In contrast, physical activities were not a predictor of global cognitive impairment. It is important to note, however, that physical activity can contribute to the development of CR, particularly when it involves cognitively stimulating activities, such as sports, hobbies, or dancing. In the present study, we only had access to data regarding physical activation of participants, without information on the cognitive aspects of these activities. This limitation may help explain our findings.

Education and leisure activities are complex activities that may promote neural connectivity both in the early years (for education) and later years (for leisure activities). In some cases, they may act as compensation processes and in other cases as neural efficiency processes during the aging process. Occupational complexity, on the other hand, seems to be more dependent on the relationship with other variables. The interaction between the different proxies of CR (both static and dynamic) reflects the complex underlying mechanisms. However, as we stated before, using a composite score of CR would have prevented us from understanding the contribution of each variable individually [2,15].

An unexpected finding was the notable proportion of participants that reverted from the cognitively impaired group to the normal group throughout the waves. This variability has been reported in recent longitudinal studies, particularly in community-based cohorts and can be explained by medical conditions, affective states, nutritional deficits, fatigue, practice effects, mistakes during the measurements, etc. [49]. Another possible explanation is the role of CR and the implementation of different lifestyle activities, which is concordant with the results of our study [49–51].

Some limitations of the present study are related to the nature of the MHAS data available, for example, for physical activity there is only one question, so we included it as a dichotomous variable, even if it has been recommended to examine doses of physical activity, such as volume, duration, frequency, or intensity [52]. Considering physical activity only as presence or absence of exercise may be a reductionist approach regarding this relevant variable. Future studies should use some scales that not only consider duration, frequency, or intensity, but also the type of physical activity performed (e.g., aerobic vs. anaerobic). Regarding the cognitive variables, one limitation of the total score used to classify the participants is that 60 out of 95 points were related to visual attention, which may bias the results toward these cognitive processes over others. However, this potential bias was partly mitigated by the use of domain-specific cognitive trajectories analyses. Another relevant limitation is the lack of measures of underlying pathology or age-related brain changes that could help us clarify the association between these variables. It would be beneficial to have a longer follow-up period to better understand the cognitive trajectories.

It is also important to highlight the strengths of the present study. One major strength is the evidence of differential effects of CR proxies on domain-specific cognitive trajectories, which let us analyze the variability of age-related impairment. Another strength is the use of a nationally representative sample provided by the MHAS, which includes both rural and urban areas of Mexico, encompassing a diverse population. This diversity enhances the generalizability of the results. Additionally, the inclusion of covariates such as sex, age, comorbidities, depressive symptoms, and socioeconomic status in the statistical analysis helped to account for factors considered as confounding variables for cognitive performance, cognitive trajectories, and risk of cognitive impairment [43]. It is also important to highlight that even with a relatively low average years of education of our sample (M = 5.76, S.D. = 4.45) we found a strong association with domain-specific cognitive trajectories and lower risk of cognitive impairment. These findings can be promising for public health interventions in low-resource settings.

Future studies should explore different types of leisure activities that may explain the variability of the findings [32,53–55] to better understand the proxies of CR. Regarding years of education, due to the information available in the MHAS, we could not include verbal intelligence, which has been considered a more robust measure of CR as it may better reflect the results of educational attainment from a qualitative and dynamic perspective [56]. We therefore encourage the incorporation of this proxy in future longitudinal studies of aging population. Additionally, these longitudinal surveys should also

incorporate questions related to reported comorbidities considered as risk factors of dementia as the cardiovascular ones described in the 2024 report of the Lancet standing Commission [57].

It is relevant to investigate the proxy measures of CR that can influence cognitive trajectories in older adults. The categorization into static and dynamic proxies allows us to distinguish between the tasks that may be executed in early years as preventive strategies (educational attainment) and the tasks that can be actively implemented by individuals even in later years (as leisure activities) which can act as intervention strategies. Moreover, the differential effects in domain-specific cognitive trajectories can provide additional information to propose more addressed strategies for healthy aging.

## Author contributions

**Conceptualization:** Juan Silva-Pereyra, Carmen García-Peña.

**Data curation:** Martina Ferrari-Díaz, Ashuin Kammar-García, Juan Silva-Pereyra.

**Formal analysis:** Martina Ferrari-Díaz, Ashuin Kammar-García.

**Funding acquisition:** Carmen García-Peña.

**Investigation:** Ashuin Kammar-García.

**Methodology:** Martina Ferrari-Díaz, Ashuin Kammar-García, Juan Silva-Pereyra, Carmen García-Peña.

**Writing – original draft:** Martina Ferrari-Díaz.

**Writing – review & editing:** Ashuin Kammar-García, Juan Silva-Pereyra, Carmen García-Peña.

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
