## [Decision Letter · Decision Letter 0]

8 Sep 2025

Dear Dr. Silva-Pereyra,

Thank you for submitting your manuscript to PLOS ONE. After careful consideration, we feel that it has merit but does not fully meet PLOS ONE’s publication criteria as it currently stands. Therefore, we invite you to submit a revised version of the manuscript that addresses the points raised during the review process.

We look forward to receiving your revised manuscript.

Kind regards,

Antoine Coutrot

Academic Editor

PLOS ONE

Journal Requirements:

[The MHAS (Mexican Health and Aging Study) is partly sponsored by the National Institutes of Health/National Institute on Aging (grant number NIH R01AG018016) and the INEGI (Instituto Nacional de Estadística y Geografía) in Mexico. Data files and documentation are for public use and available at www.MHASweb.org. The funders had no role in study design, data collection and analysis, decision to publish, or preparation of the manuscript.].

Reviewers' comments:

Reviewer's Responses to Questions

**Comments to the Author**

1. Is the manuscript technically sound, and do the data support the conclusions?

Reviewer #1: Yes

Reviewer #2: Yes

2. Has the statistical analysis been performed appropriately and rigorously?

Reviewer #1: I Don't Know

Reviewer #2: Yes

3. Have the authors made all data underlying the findings in their manuscript fully available?

Reviewer #1: Yes

Reviewer #2: Yes

4. Is the manuscript presented in an intelligible fashion and written in standard English?

Reviewer #1: Yes

Reviewer #2: Yes

Reviewer #1: Thank you for the opportunity to review this interesting and well-conducted manuscript. Your study makes a valuable contribution to the literature by examining domain-specific cognitive trajectories and risk of cognitive decline in older Mexican adults, using both static and dynamic proxies of cognitive reserve. The large, nationally representative sample and the longitudinal design over 9 years are major strengths that enhance the impact of your findings.

I have a few suggestions that I believe would further strengthen the manuscript:

About operationalization of variables: Physical activity is treated as a dichotomous variable, which limits interpretability. I recommend highlighting this limitation more explicitly in the discussion.

About Methods clarity: I consider that the choice of a -1.5 z-score cutoff for defining cognitive decline should be justified with additional references or explanation.

About Discussion: Please expand the discussion on missing CR proxies (e.g., verbal intelligence, quality of education), which could influence the interpretation of your findings. Since the average years of education in your sample are relatively low, it would be valuable to highlight the implications for populations with limited educational attainment and for public health interventions in low-resource settings. Ensure consistent use of terminology (“cognitive decline” vs. “cognitive impairment”) throughout the manuscript.

Overall, this is an important and well-prepared manuscript. With minor revisions, it will provide a strong contribution to the literature on cognitive reserve and aging.

Reviewer #2: Ferrari-Díaz et al. investigated the level of cognitive reserve and its association with cognitive functions, as well as cognitive decline across a follow-up of 9 years. The main objective of this paper is to analyse how different components of cognitive reserve, static or dynamic, influence different cognitive abilities like visual attention, verbal memory, verbal fluency.

This approach is interesting by its original combination of different variables to estimate cognitive reserve, from initial cognitive abilities (level of education) to skills preservations (physical and leisure activities). I think this approach is clinically relevant to estimate the resilience of an individual against brain ageing.

This work was conducted by an experienced team in frailty and brain age. The manuscript is easy-to-understand, well written and contributes to the scientific community

My major issues with this manuscript are :

1. In the description of the methods, especially when defining CR variables, I am having difficulty to understand how you deal with the occupational complexity for older adults. Do the retired participants keep their last occupation for data ? If authors did it that way, it may be difficult to justify this variable as dynamic component of CR. Moreover, does volunteer work, or partial-time employment have been recorded in MHAS data ?

2. In the logistic regressions investigating the risk of cognitive impairment, explained by CR variables, authors explained the creation of a composite numeric score, ranging from 0 to 95 points. If I read correctly that score, 60 points out of 95 are only calculated based on visual attention ? Why highlighting that cognitive skill more than others ? This method could focus too much on attentional skills and executive functions, influencing the findings of the association with dynamic CR

3. Which were the 12 comorbidities authors adjusted for ? I understand that they attributed one point for each comorbidity for the total score, meaning that “hypertension” and “history of cancer” are treated the same ? I think that we should consider cardiovascular risk factors and Lancet’s 2024 risk factors of Dementia as serious cofounders in cohorts of older adults, even if they have few comorbidities like in the MHAS Study.

My other questions are :

4. In abstract, last sentence indicates that increasing the years of education should delay cognitive decline, do authors really think there is fewer cognitive decline or do people start from higher CR and take more time to get MCI or dementia, whether they have a real cognitive decline ?

5. In the end of the introduction, I would use the term “cognitive decline” only, instead of “MCI or dementia” because of the lack of data concerning functional living, or aetiologic of cognitive impairment. The expression “cognitive impaired group” line 206 is also convincing.

6. In the methods, authors should explain how occupational complexity, leisure activities, socioeconomic status were expressed numerically

7. Authors explained their choice in discussion, but they were no possibility to precise the type and level of physical activity in your data ?

8. Do authors can justify the choice of -1.5SD to define cognitive impairment, instead of -2SD like majority of cognitive tests ? Is there any patient that passed from cognitive impaired group to unimpaired across the waves, by changing of decade for example ?

9. Was there a visual or auditive evaluation before passing neuropsychological tests ? That may be relevant for visual attention.

10. In Table 1, I would add a line to express the percentage of cognitively impaired participants for each wave.

11. In discussion line 290, I agree with the differences made between physical activities and leisure activities, but it’s hard for me to understand the last sentence « related to a more comprehensive healthy aging ». Maybe authors should develop further their idea.

12. Very interesting point to highlight that lower levels of education keep associated with cognition, these levels are rare in other studies focusing on higher degrees especially in WEIRD countries. Is it possible to try to categorise by diploma ? Primary, high school, college ?

13. Is there much correlation between level of education and occupationnal complexity ? Maybe there is a circular effect on theses 2 variables

**Do you want your identity to be public for this peer review?** For information about this choice, including consent withdrawal, please see our Privacy Policy

Reviewer #1: No

Reviewer #2: **Yes: ** Dr Victor GILLES, MD, MSc

---

## [Author Response · Author response to Decision Letter 1]

6 Oct 2025

Dear Dr. Antoine Coutrot,

We hope this letter finds you well. We would like to express our sincere gratitude for the valuable feedback provided by the reviewers and for the time and effort dedicated to evaluating our work.

We are writing to formally answer the review comments in the "response to reviewers" file.

Kind regards,

Martina Ferrari Díaz

Juan Silva Pereyra

---

## [Decision Letter · Decision Letter 1]

25 Nov 2025

Association of cognitive reserve with 9-year domain-specific cognitive trajectories and risk of cognitive impairment in Mexican older adults

PONE-D-25-32945R1

Dear Dr. Silva-Pereyra,

We’re pleased to inform you that your manuscript has been judged scientifically suitable for publication and will be formally accepted for publication once it meets all outstanding technical requirements.

Kind regards,

Antoine Coutrot

Academic Editor

PLOS ONE

Additional Editor Comments (optional):

Reviewers' comments:

Reviewer's Responses to Questions

**Comments to the Author**

Reviewer #2: All comments have been addressed

2. Is the manuscript technically sound, and do the data support the conclusions?

Reviewer #2: Yes

3. Has the statistical analysis been performed appropriately and rigorously?

Reviewer #2: Yes

4. Have the authors made all data underlying the findings in their manuscript fully available?

Reviewer #2: Yes

5. Is the manuscript presented in an intelligible fashion and written in standard English?

Reviewer #2: Yes

Reviewer #2: (No Response)

**Do you want your identity to be public for this peer review?** For information about this choice, including consent withdrawal, please see our Privacy Policy

Reviewer #2: **Yes: ** Dr Victor GILLES

---

## [Editor Report · Acceptance letter]

1 Dec 2025

PONE-D-25-32945R1

PLOS ONE

Dear Dr. Silva-Pereyra,

I'm pleased to inform you that your manuscript has been deemed suitable for publication in PLOS ONE. Congratulations! Your manuscript is now being handed over to our production team.

Kind regards,

on behalf of

Dr. Antoine Coutrot

Academic Editor

PLOS ONE